# Oxidative Stress and Emergence of Psychosis

**DOI:** 10.3390/antiox11101870

**Published:** 2022-09-21

**Authors:** Victoria Rambaud, Aude Marzo, Boris Chaumette

**Affiliations:** 1Institute of Psychiatry and Neuroscience of Paris, Université Paris Cité, INSERM U1266, 75014 Paris, France; 2GHU-Paris Psychiatrie et Neurosciences, 75014 Paris, France; 3Department of Psychiatry, McGill University, Montreal, QC H3A 1A1, Canada

**Keywords:** schizophrenia, oxidative stress, ultra-high risk

## Abstract

Treatment and prevention strategies for schizophrenia require knowledge about the mechanisms involved in the psychotic transition. Increasing evidence suggests a redox imbalance in schizophrenia patients. This narrative review presents an overview of the scientific literature regarding blood oxidative stress markers’ evolution in the early stages of psychosis and chronic patients. Studies investigating peripheral levels of oxidative stress in schizophrenia patients, first episode of psychosis or UHR individuals were considered. A total of 76 peer-reviewed articles published from 1991 to 2022 on PubMed and EMBASE were included. Schizophrenia patients present with increased levels of oxidative damage to lipids in the blood, and decreased levels of non-enzymatic antioxidants. Genetic studies provide evidence for altered antioxidant functions in patients. Antioxidant blood levels are decreased before psychosis onset and blood levels of oxidative stress correlate with symptoms severity in patients. Finally, adjunct treatment of antipsychotics with the antioxidant N-acetyl cysteine appears to be effective in schizophrenia patients. Further studies are required to assess its efficacy as a prevention strategy. Redox imbalance might contribute to the pathophysiology of emerging psychosis and could serve as a therapeutic target for preventive or adjunctive therapies, as well as biomarkers of disease progression.

## 1. Introduction

Schizophrenia (SZ) is a complex, multifactorial psychotic disorder, affecting 1% of the population worldwide [1]. The onset of SZ typically occurs between late adolescence and early adulthood and groups together positive, negative and cognitive symptoms. Young individuals at ultra-high risk (UHR) for psychosis can be identified, during the prodromal phase of the disease, based on the presence of attenuated or time-limited psychotic symptoms, or of a familial risk-factor, along with a drop in psychosocial functioning [2]. Despite the well-established clinical criteria required to identify UHR individuals, the risk of conversion to a first episode of psychosis (FEP) in this population reaches 25% after three years of follow-up [3].

Current treatments are mostly effective against the positive symptoms [4], while their resolution does not systematically translate into functional recovery [5]. Indeed, it appears that negative and cognitive symptoms are better predictors of functional recovery [6]. Although increasing effort is being invested in the understanding of negative symptoms, new generations of antipsychotics do not seem to make a difference in the treatment of negative symptoms [7,8]. Moreover, antipsychotics are used to attenuate impairment or suffering in UHR individuals. However, in this group of individuals, treatments with antipsychotics lead to many side effects, and even increase the risk of transitioning in some individuals [9].

Therefore, it appears that the development of more effective treatments for SZ requires a better understanding of the pathophysiology of this multi-factorial disorder. Indeed, early neurobiological changes occurring during the UHR state could play a role as predictors of the transition but also therapeutical targets for prevention strategies. The most promising window of opportunity for blocking the progression or preventing the onset of SZ, is currently around the UHR state or the FEP [10].

Amongst the multiple molecular mechanisms and neural processes which are altered in psychotic patients, a growing body of evidence suggests that oxidative stress responses are overly active in patients with SZ [11]. Studies have highlighted a dysregulation in redox metabolism during the onset of psychosis and increased oxidative damage is observed in a consistent manner in UHR individuals who subsequently develop psychosis [12]. Oxidative stress results from a shift in redox balance, that is, an accumulation of pro-oxidative factors over the antioxidant defense, leading to damage to lipids, proteins and nuclear and mitochondrial DNA (mtDNA) [13]. Reactive oxygen species (ROS) are produced as a physiological process by mitochondria, immune cells, or as necessary intermediate in enzymatic reactions, and participate in redox signaling and growth regulation [14]. They can also be produced by the brain for redox signaling [15]. The brain is particularly exposed to oxidative stress, due to its intensive neuronal activity which requires high oxygen levels and leads to a higher production of ROS [16]. Moreover, the brain contains high levels of free iron and polyunsaturated fatty acids, which are oxidizing substances and cause the neurons to be particularly vulnerable to oxidative stress [17]. In addition to the oxidative stress markers observed in the blood or brain of patients with SZ, genetic studies reveal that high-risk polymorphisms occur in genes playing a role in redox regulation [18,19,20,21].

The oxidative stress theory, as a link between the multiple molecular changes observed in psychosis, can be harnessed to identify oxidative markers of the development of psychosis. Indeed, studies have highlighted a dysregulation in redox metabolism during the onset of psychosis and increased oxidative damage is observed in a consistent manner in UHR individuals who subsequently develop psychosis [12,22].

This review is driven by the present need for understanding the pathophysiological processes involved in SZ in order to improve early treatment strategies. It provides an overview of the experimental and clinical evidence examining oxidative stress in biospecimens, including blood samples and cerebrospinal fluid (CSF), in patients with SZ. More specifically, changes in levels of enzymatic and non-enzymatic antioxidant and oxidative stress markers in the blood and the CNS of patients with SZ and UHR individuals are included.

## 2. Materials and Methods

An initial general search was performed using two databases PubMed and EMBASE. We used the keywords: (“schizophrenia” OR “psychosis”) AND (“antioxidant” OR “oxidative stress”). The initial search screened titles and abstracts only. The limits used were the date of publication (from 1991 to 2022), the species studied (humans) and the language (English). Additional records were identified through other sources.

The inclusion criteria in selecting study were:(i)articles published in peer-review journal(ii)articles published in English language(iii)patients diagnosed with SZ using standard diagnostic methods according to the Diagnostic and Statistical Manual of Mental Disorders (DSM) or the International Classification of Disease (ICD) systems(iv)studies including both patients and healthy controls cohorts

Figure 1 presents the literature searching stages and the inclusion and exclusion for each stage. Studies that solely focused on a particular oxidative stress biomarker that was not commonly investigated were excluded, along with studies recruiting patients with specific subtypes of SZ only. In addition, due to the contrasting effects that oxidative stress can have on different tissues, studies investigating the levels of oxidative stress in tissues other than the serum, the plasma, or the erythrocytes were excluded. Finally, additional records that were identified from the bibliography of selected articles or from other sources may introduce a bias in the results presented in this narrative review.

## 3. Results

A total of 120 papers were identified on PubMed and 127 on EMBASE, of which 64 were relevant research articles according to the criteria mentioned in the Methods section above and Figure 1. In addition, 12 research articles from the literature bibliography were added. The selected articles included comparative studies between healthy individuals and patients with SZ or at risk of developing the disease.

A summary of the core literature used, including 59 research articles, can be found in Table 1, Table 2 and Table 3. The number of participants in each study can be found in the Appendix A. Articles were grouped according to their results regarding the levels of oxidative stress or antioxidants defense in patients compared to healthy individuals. In addition, the clinical status, including the treatment, of the patients included in each study is mentioned, along with the sample in which biomarkers were measured.

Individuals at risk of developing SZ present decreased antioxidant defenses, except for GPx enzymatic activity (Table 1). On the other hand, amongst FEP patients, more variability is found across findings (Table 2). A larger number of studies found decreased antioxidant defenses and increased oxidative damage products in this group of patients (Table 2). Nonetheless, a study by Li et al., including 354 FEP patients, found decreased oxidative damage to lipids and increased total antioxidant status (TAS) compared to controls [23].

Overall, a large number of results have assessed oxidative stress in patients with SZ, resulting in consistent evidence about a dysfunction in antioxidant defense. Despite the heterogeneity of the findings presented, it is important to note that lipid peroxidation levels are persistently increased in the blood of chronic SZ patients, and that the TAS and the GSH blood levels are decreased (Table 3). In particular, studies with more than 150 participants recruited, found that the TAS, the GPx and SOD activity were decreased in chronic SZ patients [24,25,26,27] (Table 3, Appendix A).

**Table 1 antioxidants-11-01870-t001:** Selected peripheral biomarkers of antioxidant status and oxidative damage in unaffected FDR and UHR individuals compared to controls.

Variables	Schizophrenia	Sources	Status
Antioxidant Defense Peripheral Biomarkers			
GPx	↑	Erythrocytes [28]	Unaffected FDR [28]
	↓	Serum [29]	UHR subjects [29]
Catalase	↓	Erythrocytes [28]	Unaffected FDR [28]
SOD	↓	Erythrocytes [28], Serum [29]	Unaffected FDR [28], UHR subjects [29]
TAS	↓	Serum [30], Plasma [31]	Unaffected FDR [30,31]

GPx: Glutathione Peroxidase; SOD: Superoxide Dismutase; TAS: Total Antioxidant Status; UHR: Ultra High Risk; FDR: First-Degree Relatives.

**Table 2 antioxidants-11-01870-t002:** Selected peripheral biomarkers of antioxidant status and oxidative damage in FEP individuals compared to controls.

Variables	Schizophrenia	Sources	Status
Antioxidant Defense Peripheral Biomarkers			
GPx	↑	Erythrocytes [32,33], Serum [34]	Antipsychotic-naïve [32,33,34] and antipsychotic-treated [32,34] FEP
	↔	Erythrocytes [23], Serum [35]	Antipsychotic-naïve FEP [23,35]
	↓	Erythrocytes [36], Serum [37], Plasma [38], Whole Blood [39]	Antipsychotic-naïve FEP [36,37,38,39]
GR	↔	Erythrocytes [23]	Antipsychotic-naïve FEP [23]
Catalase	↑	Plasma [38]	Antipsychotic-naïve FEP [38]
	↓	Erythrocytes [33,36,40]	Antipsychotic-naïve [33,36] and antipsychotic-treated FEP [40]
GSH	↓	Erythrocytes [41], Serum [34,42], Plasma [32,33,43]	Antipsychotic-naïve [32,33,34], antipsychotic-free [42,43] and antipsychotic-treated [32,34,41] FEP
SOD	↑	Erythrocytes [39], Serum [44], Plasma [38,45]	Antipsychotic-naïve FEP [38,39,44,45]
	↔	Erythrocytes [33,40,46], Serum [35], Plasma [36,47]	Antipsychotic-naïve [33,35,36] and antipsychotic-treated FEP [40], Antipsychotic-treated patients with SZ [46,47]
TAS	↑	Serum [34], Plasma [38]	Antipsychotic-naïve [38] and antipsychotic-treated [34,38] FEP
	↔	Serum [37,48]	Antipsychotic-naïve FEP [37,48]
	↓	Serum [49], Plasma [32,36,41,50,51]	Antipsychotic-naïve [32,36,49,50,51] and antipsychotic-treated [32,41] FEP
**Oxidative Damage Products**			
AGEs	↑	Serum [34]	Antipsychotic-naïve and antipsychotic-treated FEP [34]
Kynurenine	↓	Serum [34]	Antipsychotic-naïve and antipsychotic-treated FEP [34]
MDA/TBARS (Lipid Peroxidation)	↑	Plasma [39,52,53]	Antipsychotic-naïve FEP [39,52,53]
	↔	Plasma [23,36,40,44]	Antipsychotic-naïve [23,36,44], and antipsychotic-treated FEP [40]
	↓	Plasma [38]	Antipsychotic-naïve FEP [38]
LOOH (Lipid Peroxidation)	↑	Plasma [32]	Antipsychotic-naïve and antipsychotic-treated FEP [32]
NO	↑	Serum [42]	Antipsychotic-free FEP [42]

GPx: Glutathione Peroxidase; GR: Glutathione Reductase; GSH: Glutathione; SOD: Superoxide Dismutase; TAS: Total Antioxidant Status; AGEs: Advanced Glycation End-products; MDA: Malondialdehyde; TBARS: Thiobarbituric Acid-Reactive Substances; FEP: First Episode of Psychosis.

**Table 3 antioxidants-11-01870-t003:** Selected peripheral and brain biomarkers of antioxidant status and oxidative damage in SZ patients compared to controls.

Variables	Schizophrenia	Sources	Status
Antioxidant Defense Biomarkers in the CNS			
GSH	↓	CSF [54], mPFC [54]	Antipsychotic-naïve patients with SZ [54]
SOD	↓	CSF [55]	Antipsychotic-naïve and antipsychotic-treated patients with SZ [55]
**Antioxidant Defense Peripheral Biomarkers**			
GPx	↑	Erythrocytes [56], Serum [34], Plasma [43]	Antipsychotic-naïve and antipsychotic-treated patients with SZ [34,56], Antipsychotic-free patients with SZ [43]
	↔	Erythrocytes [46,57,58], Serum [37], Plasma [59], Whole Blood [60]	Antipsychotic-naïve [60], antipsychotic-free [46,60] and antipsychotic-treated [37,46,57,58,59,60] patients with SZ
	↓	Erythrocytes [28,61,62,63,64,65,66], Plasma [24,25,26,67]	Antipsychotic-naïve [63,64], antipsychotic-free [66] and antipsychotic-treated [24,25,26,28,61,62,63,65,67] patients with SZ
Catalase	↑	Erythrocytes [61,62,68], Serum [69]	Antipsychotic-treated patients with SZ [61,62,68,69]
	↔	Erythrocytes [46,58,65], Plasma [24,25,26,43]	Antipsychotic-free [43,46] and antipsychotic-treated [24,25,26,46,58,65] patients with SZ
	↓	Erythrocytes [28,63,66]	Antipsychotic-naïve [63], antipsychotic-free [66] and antipsychotic-treated [28,63] patients with SZ
GSH	↓	Erythrocytes [61,64,65], Serum [34,70], Plasma [71,72], Whole Blood [73,74]	Antipsychotic-naïve [34,64] and antipsychotic-treated [34,61,65,70,71,72,73,74] patients with SZ
	↔	Erythrocytes [58]	Antipsychotic-treated patients with SZ [58]
GSSG	↑	Whole Blood [74]	Antipsychotic-treated patients with SZ [74]
SOD	↑	Erythrocytes [46,56,61,65,75], Serum [69,76,77], Plasma [45,59]	Antipsychotic-naïve [75,76], antipsychotic-free [46] and antipsychotic-treated [45,56,59,61,65,69,77] patients with SZ
	↔	Erythrocytes [46], Plasma [47]	Antipsychotic-treated patients with SZ [46,47]
	↓	Erythrocytes [28,58,60,63,64,66,78], Serum [73], Plasma [24,25,26,59,67]	Antipsychotic-naïve [60,63,64,78], antipsychotic-free [60,66] and antipsychotic-treated [24,25,26,28,58,59,60,63,67,73,78] patients with SZ
Ascorbic Acid	↓	Plasma [76,79]	Antipsychotic-naïve [76] and antipsychotic-treated [79] patients with SZ
TAS	↑	Serum [34]	Antipsychotic-naïve and antipsychotic-treated patients with SZ [34]
	↔	Serum [37,60]	Antipsychotic-naïve [60], antipsychotic-free [60] and antipsychotic-treated patients with SZ [37,60]
	↓	Plasma [27,80,81]	Antipsychotic-free [81] and antipsychotic-treated [27,80,81] patients with SZ
**ROS-producing enzymes**			
XO	↑	Plasma [59]	Antipsychotic-treated patients with SZ [59]
**Oxidative Damage Products**			
AGEs	↑	Serum [34]	Antipsychotic-naïve and antipsychotic-treated patients with SZ [34]
Kynurenine	↓	Serum [34]	Antipsychotic-naïve and antipsychotic-treated patients with SZ [34]
MDA/TBARS (Lipid Peroxidation)	↑	Erythrocytes [61,62,64,65], Serum [49,69,70,76,77], Plasma [24,25,26,43,52,56,58,59,60,67,73],	Antipsychotic-naïve [49,60,64,76], antipsychotic-free [43,60] and antipsychotic-treated [24,25,26,52,56,58,59,60,61,62,65,67,69,70,73,77] patients with SZ
	↔	Erythrocytes [60,68], Serum [47]	Antipsychotic-treated patients with SZ [47,60,68]
LOOH (Lipid Peroxidation)	↑	Plasma [66]	Antipsychotic-free patients with SZ [66]
NO	↑	Plasma [59,66], Serum [73]	Antipsychotic-free [66] and antipsychotic-treated [59,73] patients with SZ

CSF: Cerebrospinal Fluid; mPFC: medial Prefrontal Cortex; GSH: Glutathione; SOD: Superoxide Dismutase; GPx: Glutathione Peroxidase; GSSG: Glutathione disulfide; TAS: Total Antioxidant Status; ROS: Reactive Oxygen Species; XO: Xanthine Oxidase; AGEs: Advanced Glycation End-products; MDA: Malondialdehyde; TBARS: Thiobarbituric Acid-Reactive Substances; LOOH: Lipid Hydroperoxide; NO: Nitric Oxide; SZ: Schizophrenia.

## 4. Discussion

### 4.1. Evidence of the Involvement of Oxidative Stress in SZ

There is growing evidence for oxidative stress imbalance in SZ, since the early phases of the disorder, but the heterogeneity across studies must be highlighted. Depending on the type of biological factors, both replicated or mixed findings have been reported. Indeed, the blood levels of the antioxidant enzymes GPx, catalase, and SOD are found to be increased by some studies, whereas other studies found it to be unchanged or even decreased. On the other hand, findings about the blood levels of the antioxidant GSH, the TAS, and the levels of several markers of oxidative stress, such as nitric oxide (NO) and malondialdehyde (MDA), are consistent across studies. Blood levels of MDA are commonly determined as thiobarbituric acid reactive substances (TBARS) and are used as a proxy for peroxidation of membrane PUFAs. Indeed, MDA is a product of lipid peroxidation. Multiple studies have demonstrated that patients with SZ have higher blood concentration of MDA [39,69,70,76] and NO, including two meta-analyses [82,83]. One of these studies revealed the good diagnostic performance for serum MDA levels in SZ patients [69]. These findings reveal that a common pathophysiological pathway leads to oxidative stress and membrane lipid damage in patients with SZ. Therefore, the discrepancy observed in the blood levels of the antioxidant enzymes in different studies could be a result of the activation of distinct antioxidant mechanisms in response to increased concentrations of ROS. Indeed, a homeostatic regulation between the GSH and PRX antioxidant systems contributes to the prevention of neuroanatomical defects in psychotic patients exposed to trauma who present with low GPx activity [84]. It seems possible that different compensatory mechanisms activate in response to the failure or the overload of one antioxidant system. Moreover, the levels of GSH and the TAS are consistently decreased in the blood of chronic SZ and FEP patients (Table 2 and Table 3). Likewise, a meta-analysis of MRS studies of antioxidant defense in the anterior cingulate cortex (ACC) of SZ patients revealed a reduction of GSH compared to controls [85]. These findings have been replicated in several studies, revealing a strong relationship between peripheral and brain GSH levels [72,86,87,88,89]. Although blood levels of oxidative stress are important to determine potential peripheral biomarkers of SZ, levels of these markers in the CNS are necessary to understand the role played by oxidative stress in the pathophysiology of the disease. For instance, low medial prefrontal cortex (mPFC) GSH concentration was shown to correlate with high levels of GPx activity in the blood of patients but not in healthy controls, reflecting a defect in compensatory mechanisms under oxidative conditions in patients with SZ [90].

Moreover, there is genetic evidence supporting the oxidative stress theory of SZ development. Indeed, in this study, the authors showed that low GSH levels in the mPFC correlate with a trinucleotide repeat polymorphism in the gene encoding the catalytic subunit of glutamate-cysteine ligase (GCLC), the rate-limiting enzyme for GSH synthesis [90]. Notably, it was found that individuals carrying the GCLC polymorphism were at higher risk of SZ [19]. Conversely, the effects of GCLC polymorphism on ACC GSH levels were not observed in a more recent study [91]. However, this study investigated SZ patients who were non-responders to treatments and found that a higher proportion of patients with the high-risk GCLC genotype were responders to clozapine [91]. These findings suggest that SZ may arise from different pathophysiological mechanisms, and that oxidative stress is one of the mechanisms at play. Another genetic evidence of reduced antioxidant defense in SZ is the high risk polymorphism in the gene encoding for the glutathione-S-transferase (GSST1) revealed by a meta-analysis [92]. Several other gene mutations associated with the risk of developing SZ, such as *DISC1*, *PROD*, *NRG* and *DTNBP1*, lead to mitochondrial dysfunction and increased oxidative stress [93,94,95,96].

Overall, evidence from genetic and biochemical studies of protein contents and activity in SZ suggests that oxidative stress is involved in the pathophysiology of SZ. However, it is important to note that oxidative stress may also be associated with other conditions such as neurodegenerative diseases, or metabolic disorders. Likewise, several limitations to these studies must be acknowledged. First, there are substantial discrepancies across the findings from these different studies. These may be a result of several factors, including the assessment of indirect markers of oxidative stress and the variability of the sample source (plasma, serum, erythrocytes). In addition, most studies report total GSH levels and do not consider the contribution of the reduced (GSH) and oxidized (GSSG) forms of GSH. Even so, reduced GSH levels are thought to reflect 80-95% of total GSH levels.

The limited replicability of these findings is further demonstrated by the fact that several studies failed to reproduce the association found between peripheral and central GSH levels [97]. However, a proteomics analysis of the changed proteins in post-mortem brains of SZ patients and healthy individuals revealed specific alterations in mitochondrial functions and oxidative stress [98]. Additionally, studies investigating levels of antioxidants in the brain of patients consistently report decreased levels compared to controls [54,55]. Therefore, investigating reliable biomarkers in the CNS would be a relevant strategy to identify individuals at risk of developing SZ in a consistent manner. However, it is an invasive method, highlighting the need for simultaneous analysis of blood and CNS redox biomarkers.

In addition, most studies consider the effects of clinical status on oxidative stress markers. Whereas most studies compare SZ patients with controls, some divide the patients into subgroups according to the duration of the disease [61], gender [67], their smoking status [99], or the subtypes of SZ spectrum disorders [32,41,59,62,67]. These studies found significant differences in antioxidant enzyme activities between males and females [67], smokers and non-smokers [99], and the different subtypes of SZ [32,41,59,62,67]. Finally, the effects of the clinical stage of SZ and antipsychotic treatments on antioxidant systems and oxidation status must be considered carefully.

### 4.2. Oxidative Stress Biomarkers and Clinical Course of SZ

The previous section reviewed the evidence for oxidative stress in SZ patients. This section will discuss these findings considering the clinical stage of the patients in order to identify pathophysiological mechanisms that may be at play during the evolution of the disease.

Despite there being only two studies investigating individuals at risk of developing psychosis, their findings converge towards increased oxidative stress in this population. In healthy individuals with a family history of psychosis (familial high risk), TAS in the blood is decreased compared to healthy individuals without a family history of psychosis [31]. Interestingly, the authors found that oxidative stress in these healthy individuals was not influenced by negative family environmental factors [31]. In addition, during the preclinical stages of psychosis, UHR individuals present with decreased activity of antioxidant enzymes SOD and GPx compared to healthy individuals [29]. Regarding the early stages of psychosis, there are many studies which focused on FEP patients and found elevated oxidative stress and defects in antioxidant systems prior to the use of antipsychotic treatments [37,53,100,101]. Indeed, one study showed that antipsychotic-naïve FEP patients present with lower blood activity of SOD than chronic SZ patients under antipsychotic treatments [45]. Increased lipid peroxidation, in association with decreased blood levels of catalase, SOD, GPx and GSH in the blood of antipsychotic-naïve FEP patients [101], seem to indicate increased oxidative stress and defects in antioxidant systems. In addition, one meta-analysis reports lower blood TAS and catalase levels in FEP patients, which are then reversed by antipsychotic treatment [13]. In this study, the authors mention that TAS and catalase blood levels could be viewed as state-markers whereas SOD blood levels, which are decreased in both FEP and chronic medicated patients, appear to be trait markers for SZ [13].

It is important to bear in mind the heterogeneity of findings as reported in Table 1. Indeed, amongst the studies reported in this review, many present contradictory results, and one meta-analysis even reports no difference in GSH levels between chronic patients, FEP patients and healthy individuals [102]. In order to understand these discrepancies, it may be recommended to investigate how oxidative stress relates with the patient’s symptom profiles. Indeed, higher levels of oxidative stress correlate positively with the severity of symptoms assessed by the Positive and Negative Symptoms Scale (PANSS) [43,66]. On the other hand, lower levels of antioxidants correlate negatively with the severity of positive and negative symptoms [27,50,51,66,71,81], and correlate positively with global cognitive functioning [41,51]. Interestingly, electrophysiological abnormalities, such as reduced gamma responses, which are frequently observed in SZ patients, correlate with GSH levels in both patients and healthy individuals [74]. Blood GSH levels were also associated with executive functions, as measured by several neuropsychological tests, in both FEP patients and healthy individuals, despite no difference in redox markers between the groups at baseline [36].

Overall, it appears that alterations in antioxidant functions are associated with symptoms severity in patients with SZ. Decreased activity of antioxidant systems has been observed in the prodromal stage of the illness, supporting the hypothesis that oxidative stress might play a causal role in the transition to psychosis [30].

### 4.3. Link between Oxidative Stress and Current Physiopathological Hypotheses

In order to understand the pathophysiological significance of oxidative stress during the psychotic transition, this section will describe the mechanisms through which oxidative stress might be involved. The different theories of SZ pathophysiology reveal interactions between the mechanisms they describe.

The neurodevelopmental hypothesis of SZ states that interactions between genetic and environmental factors influence brain development in utero, during birth and the first years of life [103]. These neurodevelopmental abnormalities become fully expressed in the mature brain, during early adulthood [103]. According to the oxidative stress hypothesis, damage caused by oxidative stress might be the molecular basis of these changes [104].

A potential source of oxidative stress in the brain comes from auto-oxidation of excess dopamine [105]. Indeed, auto-oxidizable neurotransmitters, like dopamine or epinephrine, are present in excess in the brain, and their metabolism generates large amounts of hydrogen peroxide (H_2_O_2_) [106]. Therefore, of all the brain regions, the basal ganglia, and in particular the striatum, appear to be the most at risk of damage induced by oxidative stress due to high amount of free iron [106,107] and dopamine. Moreover, the dopaminergic theory of SZ proposes that increased dopaminergic activity in the striatum is mainly responsible for the emergence of positive psychotic symptoms [108]. On the other hand, negative symptoms such as a loss of motivation (avolition), or affective flattening, can only be partially explained by hypodopaminergy in the prefrontal cortex (PFC) [108].

Pathological alterations in cortical inhibitory circuits are increasingly studied as therapeutical targets for cognitive and negative symptoms in SZ [109]. In particular, parvalbumin GABAergic interneurons (PVI) and oligodendrocytes have a high susceptibility to oxidative stress [11]. Indeed, PVI are energy demanding for high frequency neuronal synchronization [16]. Therefore, their mitochondria produce ROS at a very high rate, and they require a functional antioxidant system. Oligodendrocytes, on the other hand, have low antioxidant levels despite their high metabolic activity, and thus are also very susceptible to oxidative stress [110]. At the pathophysiological level, PVI and oligodendrocytes’ function is altered in SZ. While oligodendrocytes ensure myelination of neurons [111], PVI are required for synchronous firing [112], and both are required for synchronous network dynamics in the brain. Connectivity alterations at the functional and structural levels in SZ have been extensively studied and are present throughout all stages of the illness [113]. There is growing evidence suggesting that PVI impairments constitute a hallmark of SZ [16,114], and are involved in the excitatory/inhibitory neuronal imbalance observed in patients [115]. Altered function of PVI neurons could also be due to *N*-methyl-D-aspartate (NMDA) receptors hypofunction, a key feature of SZ, which forms the basis of the glutamatergic hypothesis of SZ physiopathology [116]. Notably, GSH, which is an essential antioxidant and plays an important role as a scavenger of ROS in brain, is a precursor of glutamate [117]. Indeed, one study found that peripheral low GSH correlates with low glutamate in the ACC [72], therefore it appears the antioxidant activity of GSH in the brain is prioritized over its role as a precursor of glutamate [117].

Eventually, oxidative stress appears to be a convergent ‘hub’ for the different theories explaining SZ physiopathology, as reviewed by Steullet, Cabungcal [118]. Figure 2 summarizes the interactions between the different theories mentioned.

### 4.4. Re-Establishing Redox Balance

In addition to the variability observed in SZ patients, but also during pre-clinical stages, patients taking antipsychotic treatments can present with increased or decreased oxidative stress. This section discusses the effects of antipsychotic treatments on oxidative stress, along with the use of antioxidants in clinical trials for SZ patients.

First generation (typical) antipsychotics, such as haloperidol, are thought to induce higher levels of lipid peroxidation in patients than second generation (atypical) antipsychotics, such as clozapine, quetiapine and risperidone [38,70,100,119,120]. Moreover, in drug-naïve FEP patients, atypical antipsychotics reduce the levels of lipid peroxidation after six weeks of treatment [44]. Typical antipsychotics can increase the metabolism of monoamines, thus leading to more ROS being produced [121], whereas atypical antipsychotics seem to demonstrate anti-oxidative and neuroprotective effects [122]. Interestingly, first generation antipsychotics seem to be more frequently associated with side effects such as extrapyramidal symptoms, deemed to be associated with oxidative stress [123,124]. Furthermore, it was found that patients with the high risk GCLC polymorphism were more likely to respond to treatment with clozapine, which suggests that this antipsychotic drug might act on redox pathways [91]. Nonetheless, discrepancies across the findings reveal that the effects of antipsychotic drugs on redox systems are subjected to inter-individual differences. Indeed, one study found that clozapine induced higher levels of lipid peroxidation than haloperidol [77]. Other studies found no effect of antipsychotic treatments on oxidative stress [60,67,81].

Still, a recent meta-analysis revealed promising results from randomized controlled clinical trials using the antioxidant N-Acetyl Cysteine (NAC) as adjunct treatment to antipsychotics in chronic and FEP patients [125]. In particular, adjunct treatment with NAC seems to improve the negative and total PANSS scores in patients [125], along with cognitive functions such as working memory [126]. More clinical trials using NAC are currently registered, which seems to confirm the importance of this hypothesis in SZ development [127]. Adjunct treatment with the antioxidant vitamin C also proved to reduce lipid peroxidation in patients, and also decreased scores on the Brief Psychiatric Symptoms Scale (BPRS) [120]. Finally, in UHR individuals, omega-3 polyunsaturated fatty acid supplementation had no effect on vitamin E, but decreased total GSH blood levels [128]. The type of antioxidants and their effectiveness at different stages of the illness still require further investigation.

## 5. Conclusions

The scientific literature on blood levels of oxidative stress markers in SZ is marked by a high variability in the findings. Harmonization of assessment and study design should be encouraged to ensure comparability and replicability and to be able to draw definitive conclusions. Indeed, considering the association between brain and blood levels of GSH, which appears to be decreased in the blood of both UHR individuals and FEP patients, it would be an interesting biomarker to consider as part of a diagnosis. However, these findings must be interpreted with caution to prevent the physiopathological mechanisms being directly inferred from dosage. Indeed, reduced antioxidant enzymes activity could indicate a reduced need for these enzymes because of low oxidative stress levels, or a defect in the enzymes leading to high oxidative stress levels. In addition, oxidative stress levels must be assessed while considering important factors such as the gender and smoking status of the patients. Nonetheless, redox mechanisms appear to play a non-negligeable role in the early phases of psychosis, and their potential value as biomarkers remains to be explored. Redox mechanisms could also help to better understand the physiopathology of emerging SZ and might serve as therapeutic targets for preventive or adjunctive therapies.

## Figures and Tables

**Figure 1 antioxidants-11-01870-f001:**
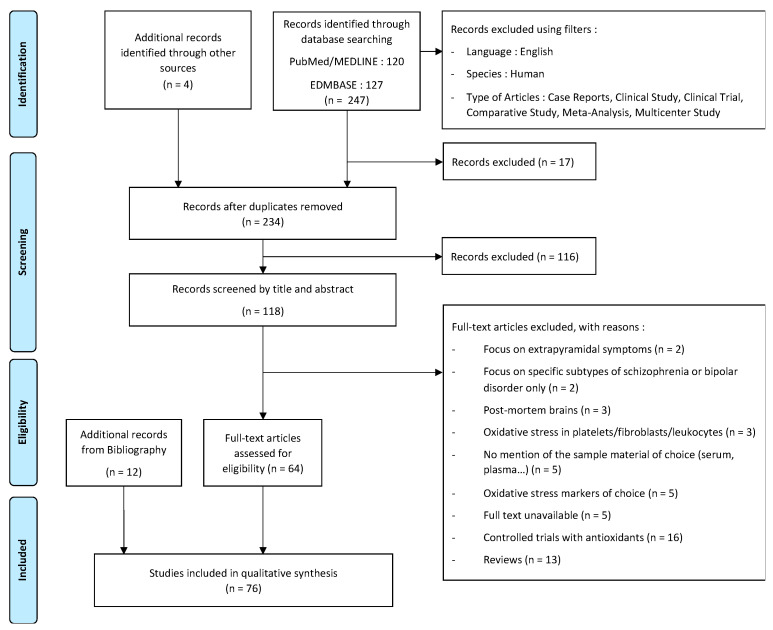
Prisma flowchart.

**Figure 2 antioxidants-11-01870-f002:**
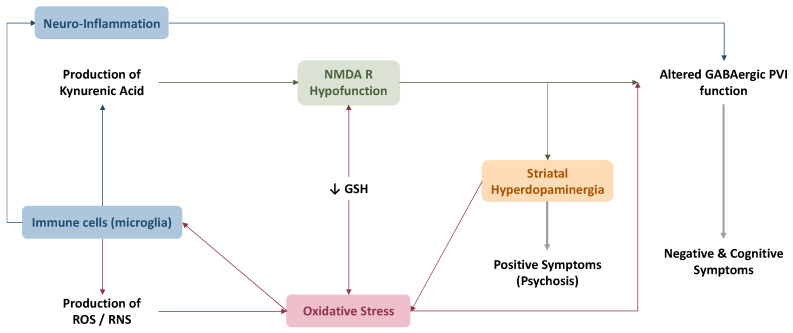
Interaction between the different theories of SZ physiopathology. Oxidative stress plays a role in the activation of immune cells which in turn lead to more oxidative stress through the production of ROS. Neuroinflammation leads to decreased activity of the GABAergic PVI, which are involved in the emergence of negative and cognitive symptoms. The release of kynurenic acid by the activated microglia acts on the NMDA receptors to decrease their function. Hypofunction of NMDA receptors is also observed when GSH levels are decreased. Overall, it leads to altered function of the GABAergic PVI, as well as increased function of dopaminergic neurons in the striatum. Striatal hyperdopaminergia is thought to contribute to the positive symptoms of SZ.

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
