# Peer review of "Oxidative Stress and Emergence of Psychosis"

_antioxidants, 2022, doi:10.3390/antiox11101870_

Round 1
Reviewer 1 Report
The manuscript is well-written and of great interest for the state of the art of biomarkers in schizophrenia.
The introduction section is well-written and well-justified. The interest and impact of the review is clearly stated. Regarding the aims there is something confuse considering the authors justified the review for improve the differenciation between UHR individividuals who transite to psychosis and those that not, but the sample include in the review is schizophrenia. Perhaps the aims should be clarified in terms of schizophrenia, although the final purpose will be to use the information as a part of early treatment.
Regarding the method section some aspects are missed. The authors should include the reference to the Prospero register. An inclusion of the risk of bias of the articles included in the review will be of help.
Figure 1 are not clear in the last block: included. The authors reviewed 64 manuscripts for eligibility and they also included 12 more. However in the box next to the full text articles the authors commented that they excluded 54 for several reasons... I think there is something wrong with the figure, it should be better review. Moreover, figure 1 should be mentioned in the text.
The results section is poor. More information about the flowchart should be of help for clarifying the figure 1. Additionally, table 1 should be better explained in the text because some information is not enough clear in the table. For instance, all the articles included in the sections found the same results regarding higher and lower levels in the articles reviewed? The information about results should be clearly explained.
The discussion section is well-written. Only one comment regarding the numbers of sub-section, in all the cases it is 1.
Author Response
We would like to sincerely thank you for your comments. We have tried to answer all the comments and modify our manuscript accordingly. Below, in the word document, are the responses to each question/comment that were raised.

Reviewer 2 Report
Very nice review on oxidative stress's impact on the etiology of SZ. Several comments are listed below in the order they first appear perusing the manuscript draft.
1. based on the chart, it seems that 54 studies of the 76 were excluded.
2. Additional discussion on tissue heterogeneity is warranted based on the sources of the redox imbalance. Surely oxidative imbalance in the blood will have a qualitatively different impact than in the brain. It would be helpful if the authors could comment on that. More importantly, it will be helpful if they can discuss whether there were studies that have simultaneously assessed oxidative stress in the blood and brain of the same subjects. As the authors have observed a discrepancy in the levels of oxidative stress in different studies, I believe it will be helpful for them also to discuss cellular heterogeneity as another critical factor that can not only explain this discrepancy but maybe provide additional insight into what cell types with oxidative stress imbalance are best predictors for detecting UHR subjects or FEP subjects.
3. While the genetic studies reported by the authors are important, one would wonder to what extent, at least on a genetic level, the oxidative stress imbalance relates to findings from the latest GWAS of SCH. There should be approximately 300 loci. Do any of these have been implicated in oxidative imbalance?
Author Response

(The authors gave the same response as above.)

Reviewer 3 Report
The review on the role of oxidative stress in the in the early stages of psychosis and chronic patients is very interesting and collects recent studies on the determination of markers of oxidative stress in schizophrenia. The article is written in an accessible manner, which helps the reader to get acquainted with the issue. Moreover, one of the strengths of the article is a broad review of the literature. In view of reports that oxidative stress plays a role in schizophrenia, I believe that this review can be published after some minor comments as presented below:
Minor comments:
please remove headings from abstract
line 58 – mitochondria produce about 90% of ROS, please mention other sources of ROS in cell
lines 64-66 needs reference
introduction – it should be mentioned that ROS also have physiological functions, not only related to pathology
references must be adjusted to the requirements of the Journal
Author Response

(The authors gave the same response as above.)

Round 2
Reviewer 1 Report
The authors have successfully review and response to my concerns.
The manuscript could be published in its present form
Author Response
Thank you very much for your helpful comments. We are glad to read that you find our answers and the revised manuscript satisfactory and worth of being published.
Sincerely yours.
Reviewer 2 Report
The authors have responded to my comments. I have no further comments.
Author Response
Thank you for your comments, they were very helpful to improve the manuscript.